# TRANSFORMED CNNS: RECASTING PRE-TRAINED CONVOLUTIONAL LAYERS WITH SELF-ATTENTION

## ABSTRACT

Vision Transformers (ViT) have recently emerged as a powerful alternative to convolutional networks (CNNs). Although hybrid models attempt to bridge the gap between these two architectures, the self-attention layers they rely on induce a strong computational bottleneck, especially at large spatial resolutions. In this work, we explore the idea of reducing the time spent training these layers by initializing them from pre-trained convolutional layers. This enables us to transition smoothly from any pre-trained CNN to its functionally identical hybrid model, called Transformed CNN (T-CNN). With only 50 epochs of fine-tuning, the resulting T-CNNs demonstrate significant performance gains over the CNN as well as substantially improved robustness. We analyze the representations learnt by the T-CNN, providing deeper insights into the fruitful interplay between convolutions and self-attention.

## INTRODUCTION

Since the success of AlexNet in 2012 (Krizhevsky et al., 2017), the field of Computer Vision has been dominated by Convolutional Neural Networks (CNNs) (LeCun et al., 1998; 1989). Their local receptive fields give them a strong inductive bias to exploit the spatial structure of natural images (Scherer et al., 2010; Schmidhuber, 2015; Goodfellow et al., 2016), while allowing them to scale to large resolutions seamlessly. Yet, this inductive bias limits their ability to capture long-range interactions.

In this regard, self-attention (SA) layers, originally introduced in language models (Bahdanau et al., 2014; Vaswani et al., 2017; Devlin et al., 2018), have gained interest as a building block for vision Ramachandran et al. (2019); Zhao et al. (2020). Recently, they gave rise to a plethora of Vision Transformer (ViT) models, able to compete with state-of-the-art CNNs in various tasks Dosovitskiy et al. (2020); Touvron et al. (2020); Wu et al. (2020); Touvron et al. (2021); Liu et al. (2021); Heo et al. (2021) while demonstrating better robustness (Bhojanapalli et al., 2021; Mao et al., 2021). However, capturing long-range dependencies necessarily comes at the cost of quadratic complexity in input size, a computational burden which many recent directions have tried to alleviate (Bello, 2021; Wang et al., 2020; Choromanski et al., 2020; Katharopoulos et al., 2020). Additionally, ViTs are generally harder to train (Zhang et al., 2019; Liu et al., 2020), and require vast amounts of pre-training (Dosovitskiy et al., 2020) or distillation from a convolutional teacher (Hinton et al., 2015; Jiang et al., 2021; Graham et al., 2021) to match the performance of CNNs.

Faced with the dilemma between efficient CNNs and powerful ViTs, several approaches have aimed to bridge the gap between these architectures. On one side, hybrid models append SA layers onto convolutional backbones (Chen et al., 2018; Bello et al., 2019; Graham et al., 2021; Chen et al., 2021; Srinivas et al., 2021), and have already fueled successful results in a variety of tasks (Carion et al., 2020; Hu et al., 2018; Chen et al., 2020; Locatello et al., 2020; Sun et al., 2019). Conversely, a line of research has studied the benefit of introducing convolutional biases in Transformer architectures to ease learning (d'Ascoli et al., 2021; Wu et al., 2021; Yuan et al., 2021). Despite these interesting compromises, modelling long-range dependencies at low computational cost remains a challenge for practitioners.

**Contributions** At a time when pre-training on vast datasets has become common practice, we ask the following question: does one need to train the SA layers during the whole learning process? Could

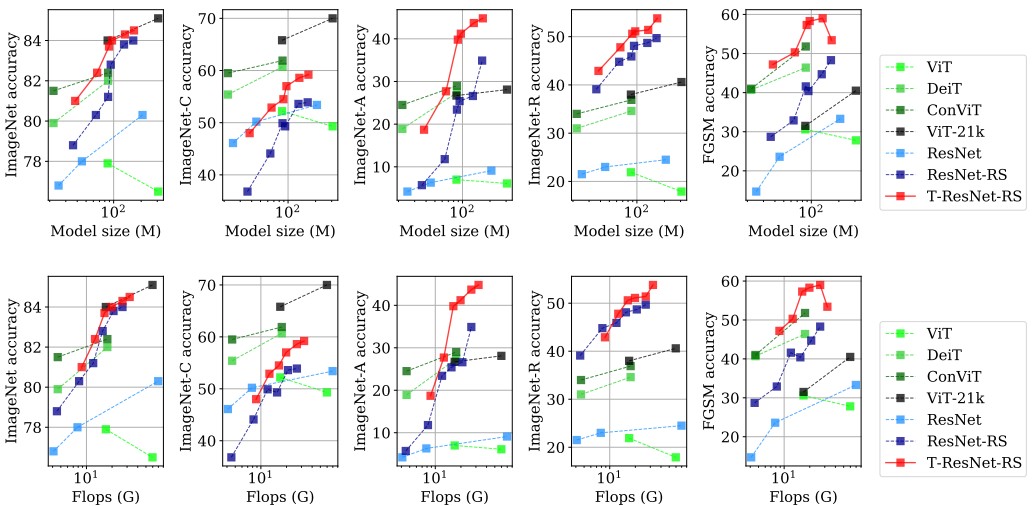

Figure 1: **Transformed ResNets strike a strong accuracy-robustness balance.** Our models (red) significantly outperform the original ResNet-RS models (dark blue) they were initialized from when evaluated on ImageNet-1k. They also exhibit solid performances on various robustness benchmarks (ImageNet-C, A and R, and FGSM adversarial attacks from left to right).

one instead learn cheap components such as convolutions first, leaving the SA layers to be learnt at the end? In this paper, we take a step in this direction by presenting a method to fully reparameterize any pre-trained convolutional layer as a *Gated Positional Self-Attention* (GPSA) layer (d'Ascoli et al., 2021). The latter is initialized to reproduce the mapping of the convolutional layer, but is then encouraged to learn more general mappings which are not accessible to the CNN by adjusting positional gating parameters.

We leverage this method to reparametrize pre-trained CNNs as functionally equivalent hybrid models. After only 50 epochs of fine-tuning, the resulting Transformed CNNs (T-CNNs) boast significant performance and robustness improvements as shown in Fig. 1, demonstrating the practical relevance of our method. We analyze the inner workings of the T-CNNs, and show that they learn more robust representations by combining convolutional heads and attentional heads in a complementary way.

**Related work** Our work mainly builds on two pillars. First, the idea that SA layers can express any convolution, introduced by Cordonnier et al. (2019). This idea was recently leveraged by d'Ascoli et al. (2021), which initialize the SA layers of end-to-end Transformers as random convolutions to imbue them with a local inductive bias and improve their sample efficiency. Our approach leverages the opposite idea: giving an end-to-end CNN the freedom to escape locality by learning self-attention at late times.

Second, we exploit the following learning paradigm: train a simple and fast model, then reparameterize it as a more complex model for the final stages of learning. This approach was studied from a scientific point of view in d'Ascoli et al. (2019), which shows that reparameterizing a CNN as a fully-connected network (FCN) halfway through training can lead the FCN to outperform the CNN. Yet, the practical relevance of this method is limited by the vast increase in number of parameters required by the FCN to functionally represent the CNN. In contrast, our reparameterization hardly increases the parameter count of the CNN, making it easily applicable to any state-of-the-art CNN. Note that these reparameterization methods can be viewed an informed version of dynamic architecture growing algorithms such as AutoGrow (Wen et al., 2020).

In the context of hybrid models, various works have studied the performance gains obtained by introducing MHSA layers in ResNets with minimal architectural changes (Srinivas et al., 2021; Graham et al., 2021; Chen et al., 2021). However, the MHSA layers used in these works are initialized randomly and need to be trained from scratch. Our approach is different, as it makes use of GPSA layers, which can be initialized to represent the same function as the convolutional layer

it replaces. We emphasize that the novelty in our work is not in the architectures used, but in the unusual way they are blended together.

## 1 BACKGROUND

**Multi-head self-attention**   The SA mechanism is based on a trainable associative memory with (key, query) vector pairs. To extract the semantic interpendencies between the $L$ elements of a sequence $\boldsymbol{X} \in \mathbb{R}^{L \times D_{in}}$, a sequence of "query" embeddings $\boldsymbol{Q} = \boldsymbol{W}_{qry}\boldsymbol{X} \in \mathbb{R}^{L \times D_h}$ is matched against another sequence of "key" embeddings $\boldsymbol{K} = \boldsymbol{W}_{key}\boldsymbol{X} \in \mathbb{R}^{L \times D_h}$ using inner products. The result is an attention matrix whose entry $(ij)$ quantifies how semantically relevant $\boldsymbol{Q}_i$ is to $\boldsymbol{K}_j$:

$$\boldsymbol{A} = \mathrm{softmax}\left(\frac{\boldsymbol{Q}\boldsymbol{K}^\top}{\sqrt{D_h}}\right) \in \mathbb{R}^{L \times L}. \tag{1}$$

Multi-head SA layers use several SA heads in parallel to allow the learning of different kinds of dependencies:

$$\mathrm{MSA}(\boldsymbol{X}) := \sum_{h=1}^{N_h} [\mathrm{SA}_h(\boldsymbol{X})]\, \boldsymbol{W}_{out}^h, \qquad \mathrm{SA}_h(\boldsymbol{X}) := \boldsymbol{A}^h \boldsymbol{X} \boldsymbol{W}_{val}^h, \tag{2}$$

where $\boldsymbol{W}_{\mathrm{val}}^h \in R^{D_{in} \times D_v}$ and $\boldsymbol{W}_{\mathrm{out}}^h \in R^{D_v \times D_{out}}$ are two learnable projections.

To incorporate positional information, ViTs usually add absolute position information to the input at embedding time, before propagating it through the SA layers. Another possibility is to replace the vanilla SA with positional SA (PSA), by including a position-dependent term in the softmax (Ramachandran et al., 2019; Shaw et al., 2018). Although there are several ways to parametrize the positional attention, we use encodings $\boldsymbol{r}_{ij}$ of the relative position of pixels $i$ and $j$ as in (Cordonnier et al., 2019):

$$\boldsymbol{A}_{ij}^h := \mathrm{softmax}\left(\boldsymbol{Q}_i^h \boldsymbol{K}_j^{h\top} + \boldsymbol{v}_{pos}^{h\top} \boldsymbol{r}_{ij}\right). \tag{3}$$

Each attention head learns an embedding $\boldsymbol{v}_{pos}^h \in \mathbb{R}^{D_{pos}}$, and the relative positional encodings $\boldsymbol{r}_{ij} \in \mathbb{R}^{D_{pos}}$ only depend on the distance between pixels $i$ and $j$, denoted denoted as a two-dimensional vector $\boldsymbol{\delta}_{ij}$.

**Self-attention as a generalized convolution**   Cordonnier et al. (2019) shows that a multi-head PSA layer (Eq. 3) with $N_h$ heads and dimension $D_{pos} \geq 3$ can express any convolutional layer of filter size $\sqrt{N_h}$, with $D_{in}$ input channels and $\min(D_v, D_{out})$ output channels, by setting the following:

$$\begin{cases} \boldsymbol{v}_{pos}^h := -\alpha^h \left(1, -2\Delta_1^h, -2\Delta_2^h, 0, \ldots 0\right) \\ \boldsymbol{r}_{\boldsymbol{\delta}} := \left(\|\boldsymbol{\delta}\|^2, \delta_1, \delta_2, 0, \ldots 0\right) \\ \boldsymbol{W}_{qry} = \boldsymbol{W}_{key} := \boldsymbol{0} \end{cases} \tag{4}$$

In the above, the *center of attention* $\boldsymbol{\Delta}^h \in \mathbb{R}^2$ is the position to which head $h$ pays most attention to, relative to the query pixel, whereas the *locality strength* $\alpha^h > 0$ determines how focused the attention is around its center $\boldsymbol{\Delta}^h$. When $\alpha^h$ is large, the attention is focused only on the pixel located at $\boldsymbol{\Delta}^h$; when $\alpha^h$ is small, the attention is spread out into a larger area. Thus, the PSA layer can achieve a convolutional attention map by setting the centers of attention $\boldsymbol{\Delta}^h$ to each of the possible positional offsets of a $\sqrt{N_h} \times \sqrt{N_h}$ convolutional kernel, and sending the locality strengths $\alpha^h$ to some large value.

## 2 APPROACH

In this section, we introduce our method for mapping a convolutional layer to a functionally equivalent PSA layer with minimal increase in parameter count. To do this, we leverage the GPSA layers introduced in d'Ascoli et al. (2021).

**Loading the filters** We want each head $h$ of the PSA layer to functionally mimic the pixel $h$ of a convolutional filter $\boldsymbol{W}_{\text{filter}} \in \mathbb{R}^{N_h \times D_{in} \times D_{out}}$, where we typically have $D_{out} \geq D_{in}$. Rewriting the action of the MHSA operator in a more explicit form, we have

$$\text{MHSA}(\boldsymbol{X}) = \sum_{h=1}^{N_h} \boldsymbol{A}^h \boldsymbol{X} \underbrace{\boldsymbol{W}_{\text{val}}^h \boldsymbol{W}_{\text{out}}^h}_{\boldsymbol{W}^h \in \mathbb{R}^{D_{in} \times D_{out}}} \tag{5}$$

In the convolutional configuration of Eq. 4, $\boldsymbol{A}^h \boldsymbol{X}$ selects pixel $h$ of $\boldsymbol{X}$. Hence, we need to set $\boldsymbol{W}^h = \boldsymbol{W}_{\text{filter}}^h$. However, as a product of matrices, the rank of $\boldsymbol{W}_h$ is bottlenecked by $D_v$. To avoid this being a limitation, we need $D_v \geq D_{in}$ (since $D_{out} \geq D_{in}$). To achieve this with a minimal number of parameters, we choose $D_v = D_{in}$, and simply set the following initialization:

$$\boldsymbol{W}_{\text{val}}^h = \boldsymbol{I}, \qquad \boldsymbol{W}_{\text{out}}^h = \boldsymbol{W}_{\text{filter}}^h. \tag{6}$$

Note that this differs from the usual choice made in SA layers, where $D_v = \lfloor D_{in}/N_h \rfloor$. However, to keep the parameter count the same, we share the same $\boldsymbol{W}_{val}^h$ across different heads $h$, since it plays a symmetric role at initialization.

Note that this reparameterization introduces three additional matrices compared to the convolutional filter: $\boldsymbol{W}_{qry}, \boldsymbol{W}_{key}, \boldsymbol{W}_{val}$, each containing $D_{in} \times D_{in}$ parameters. However, since the convolutional filter contains $N_h \times D_{in} \times D_{out}$ parameters, where we typically have $N_h = 9$ and $D_{out} \in \{D_{in}, 2D_{in}\}$, these additional matrices are much smaller than the filters and hardly increase the parameter count. This can be seen from the model sizes in Tab. 3.

**Gated Positional self-attention** Recent work (d'Ascoli et al., 2021) has highlighted an issue with standard PSA: the fact that the content and positional terms in Eq. 3 are potentially of very different magnitudes, in which case the softmax ignores the smallest of the two. This can typically lead the PSA to adopt a greedy attitude: choosing the form of attention (content or positional) which is easiest at a given time then sticking to it.

To avoid this, the authors suggest to sum the content and positional terms *after* the softmax, with their relative importances governed by a learnable *gating* parameter $\lambda_h$ (one for each attention head). The resulting Gated Positional Self-Attention (GPSA) layers are parametrized as follows:

$$\boldsymbol{A}_{ij}^h := (1 - \sigma(\lambda_h)) \, \text{softmax} \left( \boldsymbol{Q}_i^h \boldsymbol{K}_j^{h\top} \right) + \sigma(\lambda_h) \, \text{softmax} \left( \boldsymbol{v}_{pos}^{h\top} \boldsymbol{r}_{ij} \right), \tag{7}$$

where $\sigma : x \mapsto 1/(1+e^{-x})$ is the sigmoid function. In the positional part, the encodings $\boldsymbol{r}_{ij}$ are fixed rather than learnt (see Eq. 4), which makes changing input resolution straightforward (see SM. C) and leaves only 3 learnable parameters per head: $\boldsymbol{\Delta}_1, \boldsymbol{\Delta}_2$ and $\alpha$[1].

**How convolutional should the initialization be?** The convolutional initialization of GPSA layers involves two parameters, determining how strictly convolutional the behavior is: the initial value of the *locality strength* $\alpha$, which determines how focused each attention head is on its dedicated pixel, and the initial value of the *gating parameters* $\lambda$, which determines the importance of the positional information versus content. If $\lambda_h \gg 0$ and $\alpha \gg 1$, the T-CNN will perfectly reproduce the input-output function of the CNN, but may want to greedily stay in the convolutional configuration. Conversely, if $\lambda_h \ll 0$ and $\alpha \ll 1$, the T-CNN will forget about the input-output function of the CNN. Hence, we choose $\alpha = 1$ and $\lambda = 1$ to lie in between these two extremes, encouraging the T-CNNs to escape locality throughout training.

**Architectural details** To make our setup as canonical as possible, we focus on ResNet architectures (He et al., 2016), which contain 5 stages, with spatial resolution halfed and number of channels doubled at each stage. Our method involves reparameterizing $3 \times 3$ convolutions as GPSA layers with 9 attention heads. However, global SA is too costly in the first layers, where the spatial resolution is large. We therefore only reparameterize the last stage of the architecture, while replacing the first stride-2 convolution by a stride-1 convolution, exactly as in (Srinivas et al., 2021). We also add explicit padding layers to account for the padding of the original convolutions.

---

[1] Since $\alpha$ represents the temperature of the softmax, its value must stay positive at all times. To ensure this, we instead learn a rectified parameter $\tilde{\alpha}$ using the softplus function: $\alpha = \frac{1}{\beta} \log(1 + e^{-\beta\tilde{\alpha}})$, with $\beta = 5$.

Figure 2: **T-CNNs reach much higher performance and robustness at equal training time.** Total training time (original training + finetuning) is normalized by the total training time of the ResNet50-RS.

## 3   PERFORMANCE OF THE TRANSFORMED CNNs

In this section, we apply our reparametrization to state-of-the-art CNNs, then fine-tune the resulting T-CNNs to learn better representations. This method allows to fully disentangle the training of the SA layers from that of the convolutional backbone, which is of practical interest for two reasons. First, it minimizes the time spent training the SA layers, which typically have a slower throughput. Second, it separates the algorithmic choices of the CNN backbone from those of the SA layers, which are typically different; for example, CNNs are typically trained with SGD whereas SA layers perform much better with adaptive optimizers such as Adam (Zhang et al., 2019), an incompatibility which may limit the performance of usual hybrid models.

Note that our reparametrization can also be applied halfway through training: this scenario is investigated in Sec. 5. Results suggest that reparametrizing at intermediate times is optimal in terms of speed-performance trade-offs. However, for easier reproducibility, we focus on reparametrizing the fully pre-trained CNNs available in the `timm` package (Wightman, 2019): this avoids having to retrain the various models from scratch.

**Training details**   We applied our method to pre-trained ResNet-RS (Bello et al., 2021) models, using the weights provided by the `timm` package (Wightman, 2019). These models are derived from the original ResNet (He et al., 2016), but use improved architectural features and training strategies, enabling them to reach better speed-accuracy trade-offs than EfficientNets.

To minimize computational cost, we restrict our fine-tuning to 50 epochs[2]. Following (Zhang et al., 2019), we use the AdamW optimizer, with a batch size of 1024[3]. The learning rate is warmed up to $10^{-4}$ then annealed using a cosine schedule. To encourage the T-CNN to escape the convolutional configuration and learn content-based attention, we use a larger learning rate of 0.1 for the gating parameters of Eq. 7 (one could equivalently decrease the temperature of the sigmoid function).

We use the same data augmentation scheme as the DeiT (Touvron et al., 2020), as well as rather large stochastic depth coefficients $d_r$ reported in Tab. 1. Hoping that our method could be used as an alternative to the commonly used practice of fine-tuning models at higher resolution, we also increase the resolution during fine-tuning (Touvron et al., 2019). In this setting, a ResNet50 requires only 6 hours of fine-tuning on 16 V100 GPUs, compared to 33 hours for the original training. For our largest model (ResNet350-RS), the fine-tuning lasts 50 hours.

**Performance improvements**   Results are presented in Tab. 1, where we also report the baseline improvement of fine-tuning in the same setting but without SA. In all cases, our fine-tuning improves top-1 accuracy, with a significant gap over the baseline. To demonstrate the wide applicability of our method, we report similar improvements for ResNet-D architectures in SM. E.

Despite the extra fine-tuning epochs and their slower throughput, the resulting T-CNNs also outperforming the original CNNs on the ImageNet validation set at equal training budget, as shown

---

[2]We study how performance depends on the number of fine-tuning epochs in SM. D.
[3]Confirming the results of (Zhang et al., 2019), we obtained worse results with SGD.

| Backbone | Training | | | | Fine-tuning | | | | | |
|---|---|---|---|---|---|---|---|---|---|---|
| | | | | | | | Without SA | | With SA | |
| | Res. | $d_r$ | TTT | Top-1 | Res. | $d_r$ | TTT | Top-1 | TTT | Top-1 |
| ResNet50-RS | 160 | 0.0 | 1 (ref.) | 78.8 | 224 | 0.1 | 1.16 | 80.4 | 1.30 | **81.0** |
| ResNet101-RS | 192 | 0.0 | 1.39 | 80.3 | 224 | 0.1 | 1.65 | 81.9 | 1.79 | **82.4** |
| ResNet152-RS | 256 | 0.0 | 3.08 | 81.2 | 320 | 0.2 | 3.75 | 83.4 | 4.13 | **83.7** |
| ResNet200-RS | 256 | 0.1 | 4.15 | 82.8 | 320 | 0.2 | 5.04 | 83.7 | 5.42 | **84.0** |
| ResNet270-RS | 256 | 0.1 | 6.19 | 83.8 | 320 | 0.2 | 7.49 | 83.9 | 7.98 | **84.3** |
| ResNet350-RS | 288 | 0.1 | 10.49 | 84.0 | 320 | 0.2 | 12.17 | 84.1 | 12.69 | **84.5** |

Table 1: **Statistics of the models considered, trained from scratch on ImageNet.** Top-1 accuracy is measured on ImageNet-1k validation set. "TTT" stands for total training time (including fine-tuning), normalized by the total training time of the ResNet50-RS. $d_r$ is the stochastic depth coefficient used for the various models.

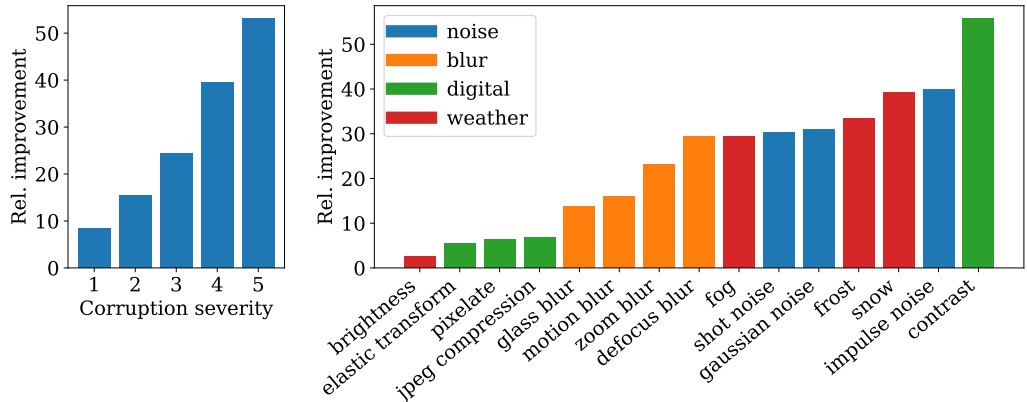

Figure 3: **Robustness is most improved for strong and blurry corruption categories.** We report the relative improvement between the top-1 accuracy of the T-ResNet50-RS and that of the ResNet50-RS on ImageNet-C, averaging over the different corruption categories (left) and corruption severities (right).

in the leftmost panel of Fig. 2[4]. However, the major benefit of the reparametrization is in terms of robustness, as shown in Fig. 2(b) and explained below.

**Robustness improvements**   Recent work has shown that Transformer-based architectures offer better robustness and out-of-domain generalization than convolutional architectures (Bhojanapalli et al., 2021; Mao et al., 2021; Mahmood et al., 2021; Shao et al., 2021). To investigate whether our fine-tuning procedure is enough to imbue CNNs with these advantages, we evaluate our T-CNNs on various benchmarks:

- **Common corruptions**: we use ImageNet-C (Hendrycks & Dietterich, 2019), a dataset containing 15 sets of randomly generated corruptions, grouped into 4 categories: 'noise', 'blur', 'weather', and 'digital'. Each corruption type has five levels of severity, resulting in 75 distinct corruptions. Note that to avoid distorting the corruptions, which are often pixel-based, we keep a resolution of 224 at inference, which disadvantages the large models trained at higher resolutions.

- **Adversarial robustness**: following (Mao et al., 2021), we evaluate the accuracy of our models under two white-bow attacks[5]: (i) single-step FGSM (Goodfellow et al., 2014) and (ii) multi-step $L_\infty$-PGD (Madry et al., 2017) with $t = 5$ steps of size $\alpha = 0.5$. Both attackers perturb the input

---

[4]We estimated the training times of the original ResNet-RS models based on their throughput, for the same hardware as used for the T-ResNet-RS.

[5]We use the toolkit provided by https://github.com/bethgelab/foolbox.

image with max magnitude $\epsilon = 1$. We also evaluate our models on ImageNet-A (Hendrycks et al., 2021), a dataset containing naturally "adversarial" examples from ImageNet. Note however that since this dataset is built from the flaws of a ResNet, it is potential unfair to CNNs.

- **Distribution shifts**: we use ImageNet-R (Hendrycks et al., 2020), a dataset with various stylized "renditions" of ImageNet images ranging from paintings to embroidery, which strongly modify the local image statistics.

The full table of results is presented in Tab. 3 of SM. A, and illustrated in Figs. 1 and 2. The T-ResNet-RS substantially outperforms the ResNet-RS on all robustness benchmarks. For example, our T-ResNet101-RS, which is 50% faster than ResNet200-RS, reaches similar or better results all robustness tasks, despite its lower top-1 accuracy on ImageNet-1k. This demonstrates that SA improves robustness more than it improves classification accuracy. The most striking improvement is in terms of adversarial robustness, where the smallest T-ResNet-RS is on par with the largest ResNet-RS despite requiring 5 times less compute.

To better understand where the benefits come from, we decompose the improvement of the T-ResNet50-RS over the various corruption severeties and categories of ImageNet-C in Fig. 3. We observe that improvement increases almost linearly with corruption severity. Although performance is higher in all corruption categories, there is a strong variability: the T-CNN shines particularly in tasks where the objects in the image are less sharp due to lack of contrast, bad weather or blurriness. We attribute this to the ability of SA to distinguish shapes in the image, as investigated in Sec 4.

## 4 DISSECTING THE TRANSFORMED CNNS

In this section, we analyze various observables to understand how the representations of a T-ResNet270-RS evolve from those of the ResNet270-RS throughout training.

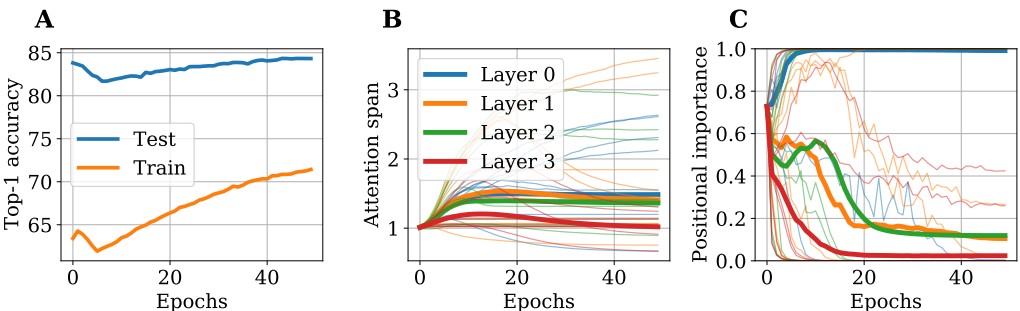

Figure 4: **The later layers effectively escape the convolutional configuration. A:** top-1 accuracy throughout the 50 epochs of fine-tuning of a T-ResNet270-RS. **B:** size of the receptive field of the various heads $h$ (thin lines), calculated as $\alpha_h^{-1}$ (see Eq. 3). Thick lines represent the average over the heads. **C:** depicts how much attention the various heads $h$ (thin lines) pay to positional information, through the value of $\sigma(\lambda_h)$ (see Eq. 7). Thick lines represent the average over the heads.

**Unlearn to better relearn** In Fig. 4A, we display the train and test accuracy throughout training[6]. The dynamics decompose into two distinct phases: accuracy dips down during the learning rate warmup phase (first 5 epochs of training), then increases back up as the learning rate is decayed.

As shown in SM. B, the depth of the dip depends on the learning rate. For too small learning rates, the dip is small, but the test accuracy increases too slowly after the dip; for too large learning rates, the test accuracy increases rapidly after the dip, but the dip is too deep to be compensated for. This suggests that the T-CNN needs to "unlearn" to some extent, a phenomenon reminiscent of the "catapult" mechanism of Lewkowycz et al. (2020) which propels models out of sharp minima to land in wider minima.

---

[6]The train accuracy is lower than the test accuracy due to the heavy data augmentation used during fine-tuning.

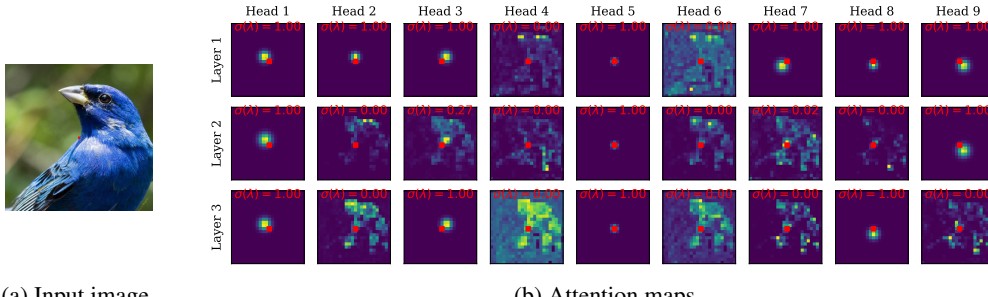

(a) Input image                                         (b) Attention maps

Figure 5: **GPSA layers combine local and global attention in a complementary way.** We depicted the attention maps of the four GPSA layers of the T-ResNet270-RS, obtained by feeding the image on the left through the convolutional backbone, then selecting a query pixel in the center of the image (red box). For each head $h$, we indicate the value of the gating parameter $\sigma(\lambda_h)$ in red (see Eq. 7). In each layer, at least one of the heads learns to perform content-based attention ($\sigma(\lambda_h) = 0$).

**Escaping the convolutional representation** In Fig. 4B, we show the evolution of the "attention span" $1/\alpha_h$ (see Eq. 4), which reflects the size of the receptive field of attention head $h$. On average (thick lines), this quantity increases in the first three layers, showing that the attention span widens, but variability exists among different attention heads (thin lines): some broaden their receptive field, whereas others contract it.

In Fig. 4C, we show the evolution of the gating parameters $\lambda^h$ of Eq. 7, which reflect how much attention head $h$ pays to position versus content. Interestingly, the first layer stays strongly convolutional on average, as $\mathbb{E}_h \sigma(\lambda_h)$ rapidly becomes close to one (thick blue line). The other layers strongly escape locality, with most attention heads focusing on content information at the end of fine-tuning.

In Fig. 5, we display the attention maps after fine-tuning. A clear divide appears between the "convolutional" attention heads, which remain close to their initialization, and the "content-based" attention heads, which learn more complex dependencies. Notice that the attention head initially focusing on the query pixel (head 5) stays convolutional in all layers. Throughout the layers, the edges of the central object is more and more clearly visible, as observed in (Caron et al., 2021). This supports the hypothesis that robustness gains obtained for blurry corruptions (see Fig. 3) are partly due to the ability of the SA layers to isolate objects from the background.

## 5 WHEN SHOULD ONE START LEARNING THE SELF-ATTENTION LAYERS?

We have demonstrated the benefits of initializing T-CNNs from pre-trained CNNs, a very compelling procedure given the wide availability of pretrained models. But one may ask: how does this compare to training a hybrid model from scratch? More generally, given a computational budget, how long should the SA layers be trained compared to the convolutional backbone?

**Transformed CNN versus hybrid models** To answer the first question, we consider a ResNet-50 trained on ImageNet for 400 epochs. We use SGD with momentum 0.9 and a batch size of 1024, warming up the learning rate for 5 epochs before a cosine decay. To achieve a strong baseline, we use the same augmentation scheme as in Touvron et al. (2020) for the DeiT. Results are reported in Tab. 2. In this modern training setting, the vanilla ResNet50 reaches a solid performance of 79.04% on ImageNet, well above the 77% usually reported in litterature.

The T-CNN obtained by fine-tuning the ResNet for 50 epochs at same resolution obtains a top-1 accuracy of 79.88%, with a 15% increase in training time, and 80.84 as resolution 320, with a 35% increase in training time. In comparison, the hybrid model trained for 400 epochs in the same setting only reaches 79.95%, in spite of a 40% increase in training time. Hence, fine-tuning yields better results than training the hybrid model from scratch.

**What is the best time to reparametrize?** We now study a scenario between the two extreme cases: what happens if we reparametrize halfway through training? To investigate this question in

| Name | $t_1$ | $t_2$ | Train time | Top-1 |
|---|---|---|---|---|
| Vanilla CNN | 400 | 0 | 2.0k mn | 79.04 |
| Vanilla CNN↑320 | 450 | 0 | 2.4k mn | **79.78** |
| T-CNN | 400 | 50 | 2.3k mn | 79.88 |
| T-CNN↑320 | 400 | 50 | 2.7k mn | **80.84** |
| Vanilla hybrid | 0 | 400 | 2.8k mn | 79.95 |
| T-CNN$^\star$ | 100 | 300 | 2.6k mn | **80.44** |
| T-CNN$^\star$ | 200 | 200 | 2.4k mn | 80.28 |
| T-CNN$^\star$ | 300 | 100 | 2.2k mn | 79.28 |

Table 2: **The benefit of late reparametrization.** We report the top-1 accuracy of a ResNet-50 on ImageNet reparameterized at various times $t_1$ during training. ↑320 stands for fine-tuning at resolution 320. The models with a $\star$ keep the same optimizer after reparametrization, in contrast with the usual T-CNNs.

a systematic way, we train the ResNet50 for $t_1$ epochs, then reparametrize and resume training for another $t_2$ epochs, ensuring that $t_1 + t_2 = 400$ in all cases. Hence, $t_1 = 400$, amounts to the vanilla ResNet50, whereas $t_1 = 0$ corresponds to the hybrid model trained from scratch. To study how final performance depends on $t_1$ in a fair setting, we keep the same optimizer and learning rate after the reparametrization, in contrast with the fine-tuning procedure which uses fresh optimizer.

Results are presented in Tab. 2. Interestingly, the final performance evolves non-monotonically, reaching a maximum of $80.44$ for $t_1 = 100$, then decreasing back down as the SA layers have less and less time to learn. This non-monotonicity is remarkably similar to that observed in d'Ascoli et al. (2019), where reparameterizing a CNN as a FCN in the early stages of training enables the FCN to outperform the CNN. Crucially, this result suggests that reparametrizing during training not only saves time, but also helps the T-CNN find better solutions.

## DISCUSSION

In this work, we showed that complex building blocks such as self-attention layers need not be trained from start. Instead, one can save in compute time while gaining in performance and robustness by initializing them from pre-trained convolutional layers. At a time where energy savings and robustness are key stakes, we believe this finding is important.

On the practical side, our fine-tuning method offers an interesting new direction for practitioners. One limitation of our method is the prohibitive cost of reparametrizing the early stages of CNNs. This cost could however be alleviated by using linear attention methods (Wang et al., 2020), an important direction for future work. Note also that while our T-CNNs significantly improve the robustness of CNNs, they do not always reach the performance of end-to-end Transformers such as the DeiT (for example on ImageNet-C, see Fig. 1). Bridging this gap is an important next step for hybrid models.

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

| Model | Res. | Params | Speed | Flops | IN-1k | IN-C | IN-A | IN-R | FGSM | PGD |
|---|---|---|---|---|---|---|---|---|---|---|
| Transformers | | | | | | | | | | |
| ViT-B/16[‡] | 224 | 86 M | 182 | 16.9 | 77.9 | 52.2 | 7.0 | 21.9 | 30.6 | 14.3 |
| ViT-L/16[‡] | 224 | 307 M | 55 | 59.7 | 76.5 | 49.3 | 6.1 | 17.9 | 27.8 | 13.0 |
| ViT-B/16 21k[‡] | 224 | 86 M | 182 | 16.9 | 84.0 | 65.8 | 26.7 | 38.0 | 31.3 | 10.3 |
| ViT-L/16 21k[‡] | 224 | 307 M | 55 | 59.7 | **85.1** | **70.0** | 28.1 | 40.6 | 40.5 | 16.2 |
| DeiT-S[†] | 224 | 22 M | 544 | 4.6 | 79.9 | 55.4 | 18.9 | 31.0 | 40.7 | 16.7 |
| DeiT-B[†] | 224 | 87 M | 182 | 17.6 | 82.0 | 60.7 | 27.4 | 34.6 | 46.4 | 21.3 |
| ConViT-S[†] | 224 | 28 M | 296 | 5.4 | 81.5 | 59.5 | 24.5 | 34.0 | 41.0 | 17.2 |
| ConViT-B[†] | 224 | 87 M | 139 | 17.7 | 82.4 | 61.9 | 29.0 | 36.9 | 51.8 | 20.8 |
| RVT-S[†] | 224 | 23.3 M | - | 4.7 | 81.9 | - | 25.7 | 47.7 | 51.8 | 28.2 |
| RVT-B[†] | 224 | 91.8 M | - | 17.7 | 82.6 | - | 28.5 | 48.7 | 53.0 | 29.9 |
| CNNs | | | | | | | | | | |
| ResNet50[‡] | 224 | 25 M | 736 | 4.1 | 76.8 | 46.1 | 4.2 | 21.5 | - | - |
| ResNet101[‡] | 224 | 45 M | 435 | 7.85 | 78.0 | 50.2 | 6.3 | 23.0 | 14.7 | 2.0 |
| ResNet101x3[‡] | 224 | 207 M | 62 | 69.6 | 80.3 | 53.4 | 9.1 | 24.5 | 23.6 | 7.3 |
| ResNet152x4[‡] | 224 | 965 M | 18 | 183.1 | 80.4 | 54.5 | 11.6 | 25.8 | 33.3 | 10.5 |
| ResNet50-RS | 160 | 36 M | 938 | 4.6 | 78.8 | 36.8 | 5.7 | 39.1 | 28.7 | 18.4 |
| ResNet101-RS | 192 | 64 M | 674 | 12.1 | 80.3 | 44.1 | 11.8 | 44.8 | 32.9 | 18.8 |
| ResNet152-RS | 256 | 87 M | 304 | 31.2 | 81.2 | 49.9 | 23.4 | 45.9 | 41.6 | 28.5 |
| ResNet200-RS | 256 | 93 M | 225 | 40.4 | 82.8 | 49.3 | 25.4 | 48.1 | 40.4 | 24.6 |
| ResNet270-RS | 256 | 130 M | 152 | 54.2 | 83.8 | 53.6 | 26.6 | 48.7 | 44.7 | 30.3 |
| ResNet350-RS | 288 | 164 M | 89 | 87.5 | 84.0 | 53.9 | 34.9 | 49.7 | 48.3 | 34.6 |
| Our Transformed CNNs | | | | | | | | | | |
| T-ResNet50-RS | 224 | 38 M | 447 | 17.6 | 81.0 | 48.0 | 18.7 | 42.9 | 47.2 | 33.9 |
| T-ResNet101-RS | 224 | 66 M | 334 | 25.1 | 82.4 | 52.9 | 27.7 | 47.8 | 50.3 | 34.2 |
| T-ResNet152-RS | 320 | 89 M | 128 | 65.8 | 83.7 | 54.5 | 39.8 | 50.6 | 57.3 | 36.8 |
| T-ResNet200-RS | 320 | 96 M | 105 | 80.2 | 84.0 | 57.0 | 41.2 | 51.1 | 58.3 | 36.4 |
| T-ResNet270-RS | 320 | 133 M | 75 | 107.2 | 84.3 | 58.6 | 43.7 | 51.4 | **59.0** | **36.6** |
| T-ResNet350-RS | 320 | 167 M | 61 | 130.5 | 84.5 | 59.2 | **44.8** | **53.8** | 53.4 | 36.4 |

Table 3: **Accuracy of our models on various benchmarks.** Throughput is the number of images processed per second on a V100 GPU at batch size 32. For ImageNet-C, we keep a resolution of 224 at test time to avoid distorting the corruptions; this disadvantages our large models, which are trained at higher resolutions. †: reported from (Mao et al., 2021) (we recalculated ImageNet-C accuracies, as the original paper reports MCE). ‡: reported from (Bhojanapalli et al., 2021) (in their setup, PGD uses 8 steps with a stepsize of 1/8).

## A    PERFORMANCE TABLE

In Tab. 3, we display the characteristics and the performance of our T-ResNet-RS models and compare them to the original ResNet-RS models as well as several other strong baselines reported in Bhojanapalli et al. (2021); Mao et al. (2021).

## B    CHANGING THE LEARNING RATE

We have shown that the learning dynamics decompose into two phases: the learning rate warmup phase, where the test loss drops, then the learning rate decay phase, where the test loss increases again. This could lead one to think that the maximal learning rate is too high, and the dip could be avoided by choosing a lower learning rate. Yet this is not the case, as shown in Fig. 6. Reducing the maximal learning rate indeed reduces the dip, but it also slows down the increase in the second phase of learning. This confirms that the model needs to "unlearn" the right amount to find better solutions.

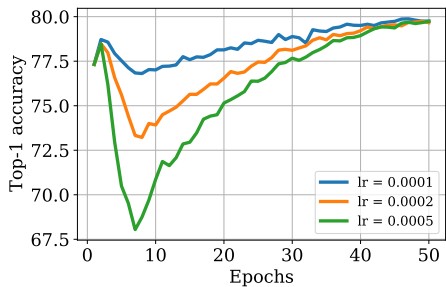

Figure 6: **The larger the learning rate, the lower the test accuracy dips, but the faster it climbs back up.** We show the dynamics of the ResNet50, fine-tuned for 50 epochs at resolution 224, for three different values of the maximal learning rate.

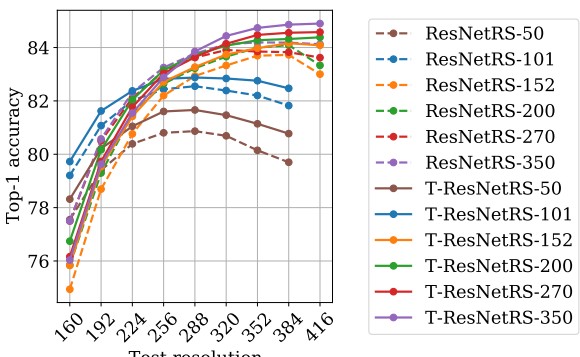

Figure 7: **Performance at different test-time resolutions, for the finetuned models with and without SA.** The ResNet50-RS and ResNet101-RS models are finetuned at resolution 224, and all other models are finetuned at resolution 320.

## C   CHANGING THE TEST RESOLUTION

One advantage of the GPSA layers introduced by d'Ascoli et al. (2021) is how easily they adapt to different image resolutions. Indeed, the positional embeddings they use are fixed rather than learnt. They simply consist in 3 values for each pair of pixels: their euclidean distance $\|\delta\|$, as well as their coordinate distance $\delta_1, \delta_2$ (see Eq. 4). Our implementation automatically adjusts these embeddings to the input image, allowing us to change the test resolution seamlessly.

In Fig. 7, we show how the top-1 accuracies of our T-ResNet-RS models compares to those of the ResNet-RS models finetuned at same resolution but without SA. At test resolution 416, our T-ResNetRS-350 reaches an impressive top-1 accuracy of 84.9%, beyond those of the best EfficientNets and BotNets Srinivas et al. (2021).

## D   CHANGING THE NUMBER OF EPOCHS

In Tab. 4, we show how the top-1 accuracy of the T-ResNet-RS model changes with the number of fine-tuning epochs. As expected, performance increases significantly as we fine-tune for longer, yet we chose to set a maximum of 50 fine-tuning epochs to keep the computational cost of fine-tuning well below that of the original training.

| Model | Epochs | Top-1 acc |
|---|---|---|
| ResNet50-RS | 0 | 79.91 |
| T-ResNet50-RS | 10 | 80.11 |
| T-ResNet50-RS | 20 | 80.51 |
| T-ResNet50-RS | 50 | **81.02** |
| ResNet101-RS | 0 | 81.70 |
| T-ResNet101-RS | 10 | 81.54 |
| T-ResNet101-RS | 20 | 81.90 |
| T-ResNet101-RS | 50 | **82.39** |

Table 4: **Longer fine-tuning increases final performance.** We report the top-1 accuracies of our models on ImageNet-1k at resolution 224.

## E CHANGING THE ARCHITECTURE

Our framework, which builds on the timm package, makes changing the original CNN architecture very easy. We applied our fine-tuning procedure to the ResNet-D models He et al. (2019) with the exact same hyperparameters, and observed substantial performance gains, similar to the ones obtained for ResNet-RS, see Tab. 5. This suggests the wide applicability of our method.

| Model | Original res. | Original acc. | Fine-tune res. | Fine-tune acc. | Gain |
|---|---|---|---|---|---|
| T-ResNet50-D | 224 | 80.6 | 320 | 81.6 | +1.0 |
| T-ResNet101-D | 320 | 82.3 | 384 | 83.1 | +0.8 |
| T-ResNet152-D | 320 | 83.1 | 384 | 83.8 | +0.7 |
| T-ResNet200-D | 320 | 83.2 | 384 | **83.9** | +0.7 |
| T-ResNet50-RS | 160 | 78.8 | 224 | 81.0 | +2.8 |
| T-ResNet101-RS | 192 | 81.2 | 224 | 82.4 | +1.2 |
| T-ResNet152-RS | 256 | 83.0 | 320 | 83.7 | +0.7 |
| T-ResNet200-RS | 256 | 83.4 | 320 | **84.0** | +0.6 |

Table 5: **Comparing the performance gains of the ResNet-RS and ResNet-D architectures.** Top-1 accuracy is measured on ImageNet-1k validation set. The pre-trained models are all taken from the timm library Wightman (2019).

