# OpenReview forum: "Transformed CNNs: recasting pre-trained convolutional layers with self-attention"
_ICLR.cc/2022/Conference — ICLR 2022 Submitted_

### Official Review · Reviewer_zV42 · 2021-11-03

**Correctness:** 4
**Technical Novelty And Significance:** 2
**Empirical Novelty And Significance:** 2
**Recommendation:** 6
**Confidence:** 4

**Main Review:**

The idea is simple and straightforward. The motivation makes sense to me and the results look good, however there are several things can be improved.
The novelty is somewhat limited, the authors did not really propose new techniques but reused components from the previous paper on a not-so-new setting.
When showing robustness improvements, the authors' motivation is that transformer-based architectures offer better robustness and out-of-domain generalization than convolutional architectures. However, from Fig.1 it seems that the Transformer-based baselines do not clearly outperform the convolutional baselines (ResNet-RS is better sometimes),  also, why there is a significant drop in the FGSM dataset when increasing model size or training time, do the authors have some explanation on these phenomenons?
Would be great if table 1 can include some transformer baseline for comparison.


**Summary Of The Paper:**

This paper shows that by leveraging a previous work Gated Positional Self-Attention layer, we can reparameterize the late stages of pre-trained CNNs and make them be hybrid Transformed CNNs. Experiments show that it can boost the performance and robustness, without training transformer block from scratch, reducing the training cost.

**Summary Of The Review:**

Overall I think the paper shows an interesting direction and present good results. However, the limited novelty prevents me from giving a higher rating.

---

### Official Review · Reviewer_ZjBY · 2021-11-03

**Correctness:** 3
**Technical Novelty And Significance:** 3
**Empirical Novelty And Significance:** 3
**Recommendation:** 6
**Confidence:** 3

**Main Review:**

### ***Pros:-***
- The paper is meticulously written and is structured well. There literature analysis is thorough and covers majority of the related works that inspired the proposed model.
- The idea explored in the paper of recasting of pre-trained CNN weights to improve training time efficiency is interesting and is well motivated. The main goal of improving train time efficiency is also validated in the experimental section.
- The experimental analysis is detailed and includes many ablation analysis experiments to analyze different aspects of the proposed model architecture.
- The experiments also show that proposed model is more robust to common corruptions, distribution shifts or adversarial perturbations than their counterparts with experiments on ImageNet-C, ImageNet-R, ImageNet-A and FGSM/PGD attack types.

### ***Cons:-***
- One of the concern includes one of the the motivation of the proposed method, which is efficient training time. Since, the training time cost is a one-time cost which incurs only while training the model, it does not seem to be an important aspect of the model. The test-time cost is a much better motivation in comparison as it has many benefits. Having efficient training is indeed an important aspect, but test-time efficiency is much more important than that and current experimental analysis does not include such analysis.
- The proposed model is not compared with many of the existing model architectures that try to create a hybrid model combining CNN and transformers in interesting manner. Furthermore, most model architecture work provide the benefits of such pre-trained model on any downstream tasks such as detection, segmentation etc, which is also missing from the experimental analysis.
- The performance comparison is provided with number of parameters vs accuracy graph, which does have its place, but additional comparison with number of FLOPs vs accuracy graph is of more importance as it also provide an important aspect of the model efficiency. I would suggest to include those graph in the main paper compared to having it in the supplementary material as a table.
- Also it seems that with increase in model capacity the performance of the proposed T-CNN has diminishing returns as CNN vs T-CNN accuracies are very close. A discussion on this should be included to explain when it is beneficial to extend CNNs to T-CNNs.









**Summary Of The Paper:**

The paper explores a hybrid type of model architectures that combines the recently popular transformer based model architectures with already well established convolutional neural networks. Specifically, the proposed hybrid model follows a two-stage training strategy where first a CNN model is trained and the pre-trained weights are used to further improve the training by recasting the weights into a new transformed CNN model termed int he paper as T-CNN. The proposed T-CNN model architecture is able to outperform its CNN counterparts and other models compared in the paper. The representations learnt by the T-CNN are also analyzed in the experimental analysis section.

**Summary Of The Review:**

Overall, the paper explores an interesting direction of research. The proposed model architecture seems to be train time efficient and improves both robustness and performance of its counterpart model. There are some concerns that need to be addressed raised in the main review, which should further clarify the technical novelty and motivation for the work.

---

### Official Review · Reviewer_Q4Pp · 2021-11-04

**Correctness:** 1
**Technical Novelty And Significance:** 3
**Empirical Novelty And Significance:** 3
**Recommendation:** 6
**Confidence:** 4

**Main Review:**

Strengths:

The paper is written very well. The authors explain their contributions very clearly, putting them in adequate context, and are upfront about the limitations. The actual methodological contribution, while somewhat incremental, could be potentially quite impactful. In addition, the paper provides some interesting analysis in terms of how the function implemented by the network changes after the self-attention fine-tuning.

Limitations:

The main limitation of the work is that the experiments are extremely limited in scope. The authors provide results for two ResNet models on ImageNet and three different Imagenet-based robustness tasks. They also provide some results on what happens if finetuning occurs during training, and study the effect of varying the learning rate, but otherwise no other ablation studies are provided. I want to emphasize that the authors justify their choices adequately and thoughtfully, but I believe that additional experiments are warranted to adequately document the potential of the approach.


**Summary Of The Paper:**

This work proposes an approach to bridge CNNs and vision transformers for image recognition. The idea is to replace the last convolutional stage of a ResNet by a self-attention layer which is initialized from the weights of the convolutional stage. The approach is shown to improve the performance of the CNN, especially in terms of robustness to adversarial examples, corruptions in the input images, and domain shift.

**Summary Of The Review:**

The paper is great, but the experiments are very limited, which I believe it puts it below the bar.
*****
After the response:

I have decided to upgrade my recommendation. Upon a more careful look through the supplementary material, the paper does include some interesting ablation studies. I think it would good to provide some experiments on a dataset different to ImageNet.

---

### Official Review · Reviewer_ojmG · 2021-11-06

**Correctness:** 3
**Technical Novelty And Significance:** 2
**Empirical Novelty And Significance:** 2
**Recommendation:** 5
**Confidence:** 2

**Main Review:**

1) The novelty of the proposed method is limited as they do not always reach the performance of end-to-end Transformers.
2) The results in Table 1 do not show a significant performance improvement. It would be better if the author could discuss situations in which this method is less effective than an end-to-end Transformer.
3) Since the authors have motioned that the proposed method can save the compute time while gaining in performance and robustness, it needs more evidence that it indeed achieves a lower computational complexity than other hybrid models.
4) Personally speaking, the organization of this paper needs to be fixed, especially the related work that needs to be reorganized.

**Summary Of The Paper:**

This paper shows that complex building blocks such as self-attention layers need not be trained from start. Instead, one can save in compute time while gaining in performance and robustness by initializing them from pre-trained convolutional layers.

**Summary Of The Review:**

See the main review.

---

### Decision · Program_Chairs · 2022-01-20

**Decision:**

Reject

**Comment:**

The paper proposes a method to accelerate training of an architectural hybrid of Transformers and CNNs: first train a CNN and then use the learned parameters to initialize a more general Transformed CNN (T-CNN) model; subsequently continue training the T-CNN.

Reviewers ratings are marginal, with three "marginally above threshold" and one "marginally below threshold".  However, no reviewer makes a compelling argument for acceptance, and all reviewers point to significant weaknesses in the work.  Reviewer ojmG: "novelty of the proposed method is limited" and "do not always reach the performance of end-to-end Transformers".  Reviewer Q4Pp: "experiments are very limited" and also (after rebuttal): "it would good to provide some experiments on a dataset different to ImageNet".  Reviewer ZjBY: "proposed model is not compared with many of the existing model architectures" and (after rebuttal): "would benefit from additional experimental analysis".  Reviewer zV42: "limited novelty prevents me from giving a higher rating".

In summary, while reviewer ratings span either side of above/below the acceptance threshold, the reviewer comments point to limited novelty and limited experimental impact.  Results appear not particularly surprising or significant: while the method provide some savings in training time, it does not seem to ultimately improve top accuracy on tasks and still lags behind the latest vision transformer architectures.  The author response did not substantially change reviewer opinion.  The AC has also taken a detailed look at the paper and does not believe the contribution to be of sufficient significance to warrant acceptance.